# Unveiling the silent threat: Investigating asymptomatic plasmodium infections in Gorgora, Ethiopia through microscopy and loop-mediated isothermal amplification

Tena Cherkos[1]*, Adane Derso[1], Banchamlak Tegegne[2], Abebe Birhanu[3], Kassahun Cherkos[4], Zufan Yiheyis Abreham[1], Banchayehu Getnet[1], Tsedenya Gebeyehu[5], Mulat Yimer[6], Tegegne Eshetu[1], Wossenseged Lemma[1], Aberham Abere[1], Yalewayker Tegegne[1], Dylan R. Pillai[7]

1 Department of Medical Parasitology, School of Biomedical and Laboratory Sciences, College of Medicine and Health Sciences, University of Gondar, Gondar, Ethiopia, 2 Amhara Public Health Institute, Bahir Dar, Ethiopia, 3 Department of Medical Microbiology, School of Biomedical and Laboratory Sciences, College of Medicine and Health Sciences, University of Gondar, Gondar, Ethiopia, 4 Department of Physiotherapy, School of Medicine, College of Medicine and Health Sciences, University of Gondar, Gondar, Ethiopia, 5 Department of Quality Assurance and Laboratory Management, School of Biomedical and Laboratory Sciences, College of Medicine and Health Sciences, University of Gondar, Gondar, Ethiopia, 6 Department of Medical Laboratory Sciences, College of Medicine and Health Sciences, Bahir Dar University, Bahir Dar, Ethiopia, 7 Department of Pathology and Laboratory Medicine, Medicine, and Microbiology, University of Calgary, Calgary, Alberta, Canada

☯ These authors contributed equally to this work.

* tenacherkos476@gmail.com

## Abstract

### Background

The asymptomatic carriers of the *Plasmodium* parasite represent a large hidden reservoir of *Plasmodium* species. These individuals can carry microscopically detectable levels and sub-microscopic levels of *Plasmodium* parasitemia. As a result, the use of clinical diagnostic methods, such as Rapid Diagnosis Tests (RDTs) and Giemsa microscopy leads to underestimation of the burden of asymptomatic malaria. Thus, the use of highly sensitive molecular methods such as loop-mediated isothermal amplification techniques (LAMP) helps to overcome these limitations and is crucial to reporting the true burden of the disease.

### Objective

This study aimed to determine the prevalence of Asymptomatic *Plasmodium* infections (APIs), and evaluate light microscopy for the detection of APIs using the LAMP method as a reference in Gorgora, Western Dembia district, Northwest Ethiopia.

### Method

A community-based cross-sectional survey was carried out from May 17 to June 30, 2023, among households located in particular kebeles at Gorgora. The study participants were

**Data availability statement:** All relevant data are within the paper and/or Supporting Information files.

**Funding:** The author(s) received no specific funding for this work.

**Competing interests:** The authors declare that they have no competing interests.

**Abbreviations:** API, Asymptomatic *Plasmodium* Infection; DBS, Dried Blood Spot; DNA, Deoxyribonucleic Acid; HH, Household; LAMP, Loop mediated isothermal amplification; RDT, Rapid Diagnostic Test; WHO, World Health Organization.

chosen using a multi-stage sampling technique. One Hundred Fifteen households and one household member from each household were selected using systematic random and simple random sampling techniques, respectively. To gather sociodemographic data, semi-structured questionnaires were used. Capillary blood samples were obtained from each study participant and screened for *Plasmodium* species infections using a LAMP kit and light microscopy. The data were entered into Epi Data version 4.6 and exported to SPSS version 25 for analysis.

## Results

The prevalence of APIs through light microscopy examination and LAMP analysis was 6.1% (95%CI: 2.5-12.1) and 11.3% (95%CI: 6.2-18.6) respectively. Using LAMP as a reference, the sensitivity and specificity of microscopy were 53.8% and 100% respectively. Microscopy examination missed six *Plasmodium* infections that were positive by LAMP analysis. A measure of agreement between LAMP and Microscopy was found to be 0.67(k=0.67).

## Conclusions

A significant proportion of APIs was found which likely act as a reservoir of transmission. This study also showed a significant number of APIs were missed by microscopy techniques. Thus, efforts to control and eliminate malaria should also consider these sub-microscopic infections.

## Introduction

Malaria is still the world's leading cause of death and morbidity even with extensive control and elimination measures in place [1]. More than 40% of people living in the world are at risk of malaria to varied degrees [2]. In Africa, more than 200 million clinical episodes and nearly one million deaths occur annually [3]. Globally, there were an estimated 249 million malaria cases and 608,000 deaths in the year 2022, which showed an increase of 5 million cases compared with 2021 [4]. During this period World Health Organization (WHO) African region carried the greatest of this global malaria burden, with 82% cases and 94% deaths followed by the WHO Southeast Asia region (10% cases and 3% deaths) [4].

It is estimated that 75% of Ethiopia's landmass is prone to malaria, and 68% of its people live in malaria-risk areas, primarily at elevations of less than 2,000 meters [5,6]. *P. falciparum* and *P. vivax* are the two most common species in Ethiopia, accounting for 60%–70% and 30%–40% of malaria infections, respectively. *P. falciparum* is the most dangerous species responsible for a large number of malaria-related deaths throughout Africa, including Ethiopia [7]. The clinical manifestation of malaria ranges from severe and complicated, mild and uncomplicated to asymptomatic infections. Severe malaria clinical features include coma, convulsions, malarial anemia, hemoglobinuria, hypoglycemia, acute pulmonary edema, acute renal failure, and circulatory collapse. Whereas fever, chills, sweats, headache, vomiting, watery diarrhea, and jaundice are uncomplicated malaria manifestations [8]. From these clinical features of malaria, severe anemia and cerebral malaria constitute the major cause of death [9]. On the other hand, repeated exposures to *Plasmodium* parasites result in the development of partial immunity for protection against further complications and lead to the formation of an asymptomatic carrier [10]. API refers to the presence of *Plasmodium* parasites in the blood

of a person who has not recently taken antimalarial medication and does not have a fever or other acute malaria symptoms [11]. As individuals with API don't exhibit signs and symptoms, they do not get clinical attention, thus representing a largely hidden reservoir of *Plasmodium* parasites, given that their continuous exposure to mosquito bites fuels the transmission cycle [10,12,13]. As a result, efforts to control and eventually eliminate malaria may be gravely threatened by asymptomatic *Plasmodium* parasite carriers [14,15].

Ethiopia has committed to stepping up efforts to eliminate malaria during the past few decades, and notable strides have been made in lowering malaria-related morbidity and mortality [16]. Ethiopia established a National Malaria Elimination Roadmap in 2020 following the Global Technical Strategy on Malaria 2016–2030, and it later launched as National Malaria Strategic Plan 2021–2025 to eliminate malaria by 2030 [16,17]. One of the nation's primary elimination strategies is the employment of traditional malaria diagnostic methods like microscopy and malaria RDTs, vector control, and artemisinin-based combination therapy (ACT) for case treatment to implement this ambitious plan [16,18]. Nevertheless, these traditional malaria diagnostic techniques missed a significant portion of low-density *Plasmodium* infections [18]. Consequently, false-negative results have the potential to postpone treatment and raise the population of infected individuals in the community [19]. Furthermore, it was estimated that 20–50% of human–mosquito transmission was caused by asymptomatic submicroscopic carriers [20–23], necessitating the use of nucleic acid amplification-based and LAMP techniques, which showed tremendous improvements in the investigation of asymptomatic infections [24,25].

Light microscopy is still the gold standard for diagnosing malaria since, when properly interpreted, a positive result indicates a *Plasmodium* infection and allows for assessment of the parasite's morphology, species differentiation, and parasitemia quantification [26–29]. Despite these advantages, light microscopy lacks the sensitivity to be applied in situations where low-density parasite carriage occurs [30]. Polymerase chain reaction (PCR), a very sensitive molecular technique, is still the gold standard for diagnosing submicroscopic malaria parasites [31]. However, it is less useful for malaria mass screening programs in communities with low resources due to the high prices of sophisticated equipment like the thermal cycler, the time-consuming procedure that delays the release of results, and the requirement for skilled laboratory personnel [32]. Thus, Rapid molecular tools that are easy to use and deploy in the field, like LAMP, can be utilized in place of PCR to support malaria control and prevention programs [33,34]. Utilizing strand displacement polymerases, like *Bacillus stearothermophilus* (Bst) polymerase, LAMP is a nucleic acid amplification technique that amplifies target sequences at isothermal temperatures of about 65°C [35]. This technique eliminates the non-specific amplification of closely related nucleic acid sequences by using four or six oligonucleotide primers that recognize six or eight different sequence portions of the target [36]. LAMP is designed to detect the genus *Plasmodium* to species level with a limit of detection of as low as 5 parasites/μl of whole blood within an hour of total turnaround time [37].

Most of the studies conducted on asymptomatic malaria infection prevalence in Ethiopia were done by microscopy and RDT. Thus, malaria prevalence mostly focuses on microscopic APIs since submicroscopic infections remain undetected by those methods. Therefore, this study aimed to detect APIs using microscopy and LAMP to identify both microscopic and submicroscopic malaria infection prevalence and to evaluate light microscopy for detecting APIs using LAMP as a reference among community residents in the study area.

## Materials and methods

### Study area

This study was conducted in Gorgora, which is found in the Western Dembia district and near Lake Tana, Amhara National Regional State, Ethiopia. It is located 65 kilometers from Gondar

town and 808 kilometers from Addis Ababa, the capital city of Ethiopia [38]. Gorgora is a moist Weina Dega, rainfall ranges between 900 and 1400 mm, and the altitude is within the range of 1500 and 2300 m. The temperature in Gorgora varies from 24 to 15°c [38]. It has a latitude and longitude of 12°14′N and 37°18′E respectively. The area has an estimated total population size of 16,270 [38]. The area has ongoing malaria transmission, with an estimated prevalence of 30.3%. *Plasmodium falciparum* and *P. vivax* are the most prevalent *Plasmodium* species in the area [39].

## Study design and period

A community-based cross-sectional study was conducted to assess the prevalence of APIs using microscopy and LAMP methods and to evaluate light microscopy for detecting APIs using LAMP as a reference among community dwellers in Gorgora, Western Dembia district, Northwest Ethiopia from May 17 to June 30, 2023.

## Population

**Source population.** All individuals who are residents of Gorgora, western Dembia district, Northwest Ethiopia.

**Study population.** All individuals within the selected households (HH) who were found in each selected kebeles being available during the data collection period and fulfilled the inclusion criteria.

**Inclusion criteria.** Individuals who were permanent residents of the area, who lived at least 6 months or older, who had no signs and symptoms of malaria, whose axillary temperature is < 37.5°C and there has been no history of fever in the past 72 hours, who was not taking antimalarial treatment 1 month before and during the data collection period and who gave informed written consent to take part in the study were included in the study.

**Exclusion criteria.** Study participants who were severely ill and unable to respond and unable to provide sufficient blood samples for different reasons were not included.

## Sample size and sampling technique

**Sample size determination.** The required sample size is calculated using the single-population proportion formula considering a 95% confidence level, a design effect of 2, a margin of error of 5%, and an asymptomatic *Plasmodium* prevalence of 3.5%, from a previous study [40]. By considering a 10% non-response rate, the required sample size was calculated as follows:

$$n = \left(Z\alpha/2\right)2P\left(1 - P\right) \times DeEf/d2)$$

$$n = \left(1.96\right)2 \times \left(0.035\right) \times \left(0.965\right)/\left(0.05\right)2 \times 2$$

$$n = 103.8 + \left(103.8 \times 0.1\right)$$

$$n = 115$$

where; n is the required sample size,

Zα/2 = the value under the standard normal table for the given value of confidence level = 1.96

P = prevalence of asymptomatic malaria from the previous study.

d = margin of error

DeEf = design effect

**Sampling technique.** A multi-stage sampling technique was used to select the study participants. In the first stage, among the kebeles in Gorgora, two kebeles (danawawa and abrjiha) were selected by using a simple random sampling technique (lottery method). At the second sampling stage, the number of HHs in each kebele was determined by proportional allocation according to the total numbers of HHs in each kebele, this number of HHs was obtained from the health post. Then, HHs were selected using a systematic random sampling technique with an interval of "K" of nine. The sampling interval was calculated by dividing the total number of HHs by the number of HHs to be included in the sample for each kebele. The initial HH was randomly selected by lottery method and the next HH was selected at that interval. One study participant was selected from each HH regardless of family size by using a lottery method. In case no eligible participant was identified in a selected HH, the next HH was selected keeping the interval constant afterward.

## Data collection and laboratory methods

**Socio-demographic data.** The socio-demographic characteristics of the study participants were developed in the English language and translated into the local language (Amharic). Before starting actual data collection, training was given to data collectors about the objective of the study, study participant recruitment techniques among household members, the data collection instrument, data collection techniques, and other ethical issues by the principal investigator of this study. Then, socio-demographic data were collected by trained data collectors via face-to-face interview technique parallel with capillary blood sample collection (S1 File).

**Blood sample collection, blood film preparation, and parasite detection.** Before capillary blood sample collection, the inclusion criteria used to enroll the study participants such as axillary temperature, malaria signs, and symptoms such as headache, chills, myalgia, dizziness, nausea, and diarrhea were screened by a nurse professional with the qualification of a BSc degree. After obtaining written informed consent from the study participants who fulfilled the inclusion criteria, capillary blood samples were collected aseptically from finger pricks, using sterile blood lancets by two trained laboratory technologists. The first drop of blood was removed and consecutive drops were taken for blood film preparation and to collect a dried blood spot (DBS) in Whatman filter papers. Both thin and thick blood smears were prepared on a single slide for each of the study participants by dropping approximately 2-3 μl and 6 μl blood respectively. Each blood smear was air-dried and the thin smear was fixed by carefully dropping methanol using a Pasteur pipette.

The methanol-fixed thin smears were allowed to dry completely in the air by placing the slides on a flat surface. Then, the air-dried blood smears were transported to the nearby health center in a slide box and stained with 10% Giemsa for 10 minutes [27]. Finally, microscopic slides were transported to and examined under oil emulsion (100 X) objective at the University of Gondar medical parasitology laboratory to detect and identify *Plasmodium* parasites. A slide was considered negative if no *Plasmodium* parasite was detected after examination of at least a hundred fields of the thick smear with 100X objective [41].

For molecular detection, by applying further pressure on the capillary puncture, capillary blood was dropped into filter papers (Whatman 3MM filter papers) and separately placed DBS were collected. The filter papers were allowed to air dry and were put separately in clean Zip-lock plastic bags with silica desiccant. Then, Filter papers in zip-lock plastic bags were packed with large plastic bags and transported to the nearby health center for storage at room temperature [42,43]. Later, it was transported to Amhara Public Health Institute, Bahir Dar, Ethiopia for molecular analysis.

## Loop-mediated isothermal amplification (LAMP assay)

LAMP technology used in this study was the Humaloopamp machine (Human Gesellschaft fur Biochemica and Diagnosticambh, Wiesbaden, Germany) with its specific kits used for DNA extraction and amplification as described elsewhere [35,44]. The machine has a heating unit, amplification unit, and detection unit. The workflow consists of four main steps; sample transfer and lysis step, Loopamp PURE DNA extraction step, Loop-mediated isothermal amplification step, and result reading steps. To transfer blood samples from DBS, two 3-mm punches of the DBS were punched out using a sterile paper puncher and placed into a 1.5-mL Eppendorf tube and then treated with 100 μL nuclease-free water overnight. Thirty μl of blood sample and 30μLof normal saline were pipetted into the heating tube containing sodium hydroxide solution. For negative control, 30μl negative control supplied in the kit was pipetted into another heating tube. The heating tubes were put into the heating block at 75°C for 5 minutes to start sample lysis. DNA was eluted using the Loopamp™ PURE DNA Extraction Kit (Eiken Chemical Co. Ltd) according to the manufacturer's supplied protocol as reported elsewhere [45,46]. By putting injection caps onto the cap of the adsorbent tube and screwing them on the adsorbent tube, DNA was injected into the reaction tubes [46]. To start DNA amplification, the reaction tubes were put into the amplification unit of Huma Loop M with a temperature of 65°C and start isothermal amplification which takes 40 minutes, and subsequent inactivation at 80°C for 5 minutes. After completing the LAMP reaction, the reaction tubes were put into the fluorescent detection unit of Huma Loop M, and the Ultra Violate (UV) light was turned on to read the results [47–49]. The sample was considered positive when it fluoresced green and negative when the sample did not fluoresce green (no fluorescent formed) [46]. DNA samples were first tested using a Pan LAMP assay detecting all human-infecting *Plasmodium* spp. After this, all the positive samples were further tested using a *P. falciparum*-specific LAMP assay (Pf LAMP kit) and a *P. vivax*-specific LAMP assay (Pv LAMP kit).

## Quality control and management

Quality control for working equipment and reagents was ensured using standard controls. The quality of the Geimsa solution was ensured by checking expired dates for stock, filtering the Geimsa working solution, and using known positive and negative control slides. All microscopic slides were read by two experienced laboratory technologists independently. The discordant results were reexamination by a third trained and certified laboratory technologist blind to the initial examination results.

The quality of the LAMP reaction was ensured by including positive and negative controls in each run. Two laboratory experts read the result using UV light on the detection unit. Discordant readings were read by a third laboratory expert who was blinded to the first result readings. These laboratory experts who read the LAMP assay were blinded to the microscopic examination results.

**Ethical approval.** The study was approved by the ethical review committee of the School of Biomedical and Laboratory Science, College of Medicine and Health Sciences, University of Gondar (with protocol number Ref. No. SBMLS/5/16), Permission was also obtained from Western Dembia district health office and kebele administrators where the data were collected. Besides, written informed consent was obtained from the study participants aged ≥ 18 years old, and for participants aged < 18 years old, written informed assent was obtained from their parents or guardians.

## Data processing and analysis

Data were coded, entered into Epidata version 4.6, cleaned up, and analyzed using SPSS for Windows version 25 (IBM SPSS Statistics 25). Frequencies and summary statistics, such as

mean, standard deviation, and percentages, were generated to describe the study participants in terms of the relevant variables. Microscopy was compared with LAMP by calculating Kappa values with 95% confidence intervals to assess the agreements between tests. Sensitivity [true positive/ (true positive + false negative)], specificity [true negative/ (true negative+false positive)], and predictive values with their exact 95% confidence interval (CI) were estimated using LAMP as reference. Finally, the results of this study were presented in texts, tables, and figures accordingly.

## Results

### Socio-demographic characteristics of study participants

A total of 115 asymptomatic individuals from two kebeles participated in this study. Of the total study participants, females accounted for 67.0% (77/115) and 33.0% (38/115) were males. The mean age of study participants was 20.2 years (±10.9 year standard deviation). A majority (55.7%, 64/115) of the study participants had a previous history of malaria (Table 1).

### Asymptomatic *Plasmodium* infections detected by microscopy and LAMP methods

In this study, capillary blood samples collected from 115 asymptomatic study participants were processed, and tested for the presence of API through light microscopy examination and LAMP analysis, yielding a prevalence of 6.1% (7/115) (95%CI: 2.5-12.1) and 11.3%(13/115) (95%CI: 6.2-18.6) respectively (S1 Data Set). Of those positives by microscopy, 71.4% (5/7) and 28.6% (2/7) had *P. falciparum* and *P. vivax* infections, respectively. The LAMP method identified 13 positives from a total of 115 samples tested using Loopamp™ malaria Pan Detection kits (11.3%). From those positives, 53.8% (7/13) tested positive for *P. falciparum* using Loopamp™ malaria Pf detection kits, 30.8% (4/13) tested positive for *P. vivax* using Loopamp™ malaria Pv detection kits, and 15.4 (2/13) were mixed infections (*P. falciparum + P. vivax)* (Fig 1).

**Table 1. Socio-demographic characteristics of study participants in Gorgora, Western Dembia district, Northwest Ethiopia, May 17 to June 30, 2023.**

| Variables | Category | Frequency | Percentage |
|---|---|---|---|
| Age | 5–14 | 41 | 35.7% |
| | 15–29 | 48 | 41.7% |
| | >29 | 26 | 22.6% |
| Sex | Male | 38 | 33.0% |
| | Female | 77 | 67.0% |
| Family size | ≤ 5 | 37 | 32.2% |
| | >5 | 78 | 67.8% |
| Educational status | Illiterate | 58 | 50.4 |
| | Elementary | 40 | 34.8 |
| | High school and above | 17 | 14.8% |
| Utilization of bed net | daily using | 25 | 21.3% |
| | Some times | 31 | 27.0% |
| | Not using | 59 | 51.3% |
| Previous history of malaria | Yes | 64 | 55.7% |
| | No | 51 | 44.3% |
| Family history of malaria | Yes | 64 | 55.7% |
| | No | 51 | 44.3% |

### Performance of microscopy for detection of APIs using LAMP as reference method

The performance of microscopy was assessed using LAMP as a reference method. Out of 13 LAMP-confirmed APIs, microscopy only detected 7 of them. There were 6 positive results by LAMP but negative by microscopy. The sensitivity, specificity, positive predictive value, and negative predictive value of microscopy were 53.8%, 100%, 100%, and 94.4%, respectively. A measure of agreement (k = 0.67) was observed between microscopy and LAMP (Table 2).

## Discussion

To reduce the spread, control, and eventually eliminate malaria, it is crucial to identify *Plasmodium* parasites in asymptomatic individuals. However, in resource-limited settings especially at the community level, diagnosing asymptomatic malaria is challenging due to low parasitemia levels and a lack of more sensitive diagnostic techniques [50]. At this time, even with appropriate treatment for symptomatic cases, asymptomatic individuals continue to be potential carriers and transmitters of gametocytes, resulting in ongoing malaria transmission within a population [51,52]. The present results also indicated in the study area where treating symptomatic cases alone did not stop or reduce malaria transmission as it was expected. The considerable prevalence of asymptomatic malaria obtained by this study could be the source of forward transmission. Thus, addressing the problem associated with asymptomatic carriers is required in the study area or elsewhere to achieve the malaria elimination goals.

A robust diagnostic capacity is necessary for the timely identification and treatment of any parasitemia individuals to control and eliminate malaria [53]. Therefore, the detection of both

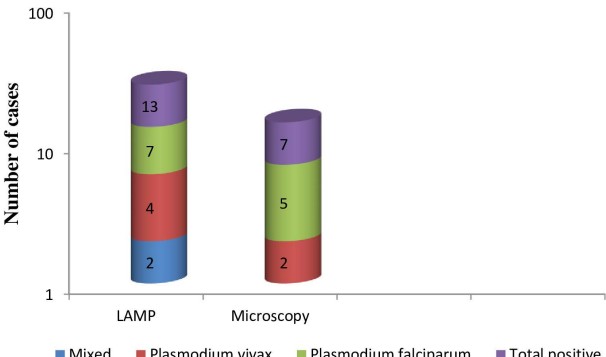

**Fig 1. Asymptomatic *plasmodium* infections in Gorgora, Northwest Ethiopia, from May 17 to June 30, 2023.**

**Table 2. Diagnostic accuracy of microscopy to detect asymptomatic *Plasmodium* infections using LAMP as a reference among community residents in Gorgora from May 17 to June 30, 2023, (N = 115).**

| Diagnostic Methods | | LAMP (used as a reference method) | | | Sensitivity (95% Cl) | Specificity (95% Cl) | PPV (95% Cl) | NPV (95% Cl) | K |
|---|---|---|---|---|---|---|---|---|---|
| | | Pos | Neg | Total | | | | | |
| Microscopy | Pos | 7 | 0 | 7 | 53.8(25.1–80.8) | 100(96.5–100) | 100(59–100) | 94.4(90.4–96.8) | 0.67 |
| | Neg | 6 | 102 | 108 | | | | | |
| | Total | 13 | 102 | 115 | | | | | |

PPV= Positive predictive value NPV= Negative predictive value, LAMP= Loop-mediated isothermal amplification, K= Kappa value, Pos= Positive, Neg= Negative.

microscopic and submicroscopic APIs is crucial to scaling up malaria control and elimination efforts.

Malaria remains a leading cause of morbidity and mortality particularly in developing nations [54]. In the current study, the rate of malaria positivity using microscopy was 6.1% (95%CI: 2.5-12.1). This finding was consistent with studies conducted in Rayakobo, Northeast Ethiopia (5.2%) [55], and Boricha District, Sidama Region, Ethiopia (4%) [56]. The LAMP analysis finding (11.3%, 95% CI: 6.2-18.6) was in line with the study conducted in Zanzibar (10.5%) [30].

On the contrary, the current study was lower than the study conducted in Metema district, Northwest Ethiopia which had (17.5%) positivity by microscopy [57], and Gambela, Southwest Ethiopia (22.2%) positivity by LAMP [52]. On the other hand, the present study was higher than studies conducted in Namibia, Zambezi region, which reported a 2.2% prevalence by LAMP [58], and Northwestern Thailand which showed a prevalence of 0.27% and 2.33% by microscopy and LAMP respectively [33]. The seasonality of transmission levels in Ethiopia could be the cause of the disparity in positivity. The district, the amount of malaria transmission intensity, the season, the density of parasitemia, the level of immunity developed, and the use of malaria chemoprophylaxis can all affect the rate of malaria positivity [56,59,60].

The present study showed that six samples that were negative by microscopy ended up positive by LAMP assay. This finding was consistent with a previous study conducted in North America that showed seven malaria cases identified by LAMP but negative by microscopy [61]. This finding might be because the majority of APIs have a parasitemia level below the detection limit of microscopy [51]. This was supported by a study from Guniea which revealed that a lower density of malaria parasitemia is highly associated with the negative result of microscopy [19]. The failure of microscopy to detect a substantial proportion of sub-microscopic individuals could also be due to low parasite density or operational shortcomings by the laboratory technician. As a result, these untreated individuals (false negative) pose a considerable challenge to the malaria control effort as they subsequently act as reservoirs of infection for continuous transmission [50]. Furthermore, in endemic areas, especially in resource-limited settings, where microscopy remains the malaria diagnostic method, submicroscopic API leaves untreated and can progress to severe and life-threatening forms of the disease [62].

Accurate diagnostic instruments are essential for diagnosing and treating malaria infections, particularly in regions where the detection of low-density and asymptomatic malaria infections is a significant challenge and when the goal is to eliminate the disease [44]. In addition to diagnostic method accuracy, factors such as affordability, the number of tests to be conducted, the equipment needed, and the availability of trained people should be taken into account when choosing diagnostic tests [63]. Asymptomatic infections are indeed often sub-patent and fall under the threshold of detection of the usual diagnostic for malaria: microscopy and rapid RDT [64].

Since then, several nucleic acid-based detection techniques have been developed, with an average detection limit range of about 1-5 P/μL. These techniques include nested PCR, multiplex PCR, and real-time PCR [65,66]. For most field applications or routine point-of-care services in most healthcare facilities, the multi-step protocols for PCR-based methods are not practical. With reliable results and greater field application adaptability, rapid target DNA detection by LAMP is a promising malaria detection technique [31,67]. Because LAMP kits use a strong closed system and never open tubes containing amplified products, there is less chance of DNA contamination than with conventional or nested PCR [35]. The LAMP accumulates approximately 109 copies of target DNA within a time frame of less than an hour [34].

Previous studies showed that microscopy had lower sensitivity than LAMP [68,69]. In the present study, Compared to LAMP, microscopy had a sensitivity and specificity of 53.8% and 100% respectively. This finding was consistent with previous studies conducted in Ethiopia that showed 55.6% sensitivity and 100% specificity [70] and systematic review and meta-analysis in Ethiopia that showed 56% sensitivity and 100% specificity compared to LAMP as a reference standard [63].

In regions where co-endemicity of non-falciparum species exists, species identification of malaria is crucial [71]. Compared to microscopy, molecular techniques can identify up to eight times as many *Plasmodium* species infections, of which up to one-third are mixed infections [34]. Using microscopy, mixed infections may be overlooked when one species is present at low parasitemia, thus clinically or epidemiologically important interactions between species may be missed [72]. In the present study, no mixed infections were detected by microscopic examination. However, two blood samples were tested positive for mixed *plasmodium* infections by LAMP analysis.

## Limitations of the study

Due to limited resources, the study was unable to utilize a more sensitive molecular tool PCR as a reference to evaluate individual diagnosis methods used in this study for detecting APIs. Not using RDTs in comparison was another limitation of this study due to resource constraints.

## Conclusions

The study showed that APIs remain a major public health challenge in the study area. This large proportion of APIs likely act as a reservoir of transmission. This has major implications for ongoing malaria control programs that are based on the treatment of symptomatic patients and highlight the need for intervention strategies targeting asymptomatic carriers. This study also showed a significant number of APIs were missed by microscopy techniques. Therefore, efforts to control and eliminate malaria should also consider these sub-microscopic infections. A further study that evaluates the performance of individual LAMP kits for the detection of APIs using PCR as a reference and its affordability is recommended to use these kits in malaria surveillance programs.

## Supporting information

**S1 File. Questionaire.**
(DOCX)

**S1 Data Set. Data entry-1.**
(XLSX)

## Acknowledgments

We would like to express our sincere gratitude to the data collectors who played a crucial role in gathering the information that forms the foundation of this work. Their dedication and meticulousness were instrumental in ensuring the quality and integrity of our data. Most importantly, we would like to express our heartfelt appreciation to the study participants. Their willingness to share their time and insights is what truly made this research possible. We are humbled by their trust and generosity, and we recognize that their participation is the cornerstone of this project's success.

## Author contributions

**Conceptualization:** Tena Cherkos, Adane Derso, Abebe Birhanu, Kassahun Cherkos, Wossenseged Lemma, Aberham Abere, Yalewayker Tegegne, Dylan R. Pillai.

**Data curation:** Tena Cherkos, Adane Derso, Banchamlak Tegegne, Abebe Birhanu, Kassahun Cherkos, Zufan Yiheyis Abreham, Banchayehu Getnet, Tsedenya Gebeyehu, Mulat Yimer, Tegegne Eshetu, Wossenseged Lemma, Aberham Abere, Yalewayker Tegegne, Dylan R. Pillai.

**Formal analysis:** Tena Cherkos, Adane Derso, Banchamlak Tegegne, Abebe Birhanu, Kassahun Cherkos, Mulat Yimer, Wossenseged Lemma, Aberham Abere, Yalewayker Tegegne, Dylan R. Pillai.

**Investigation:** Tena Cherkos, Adane Derso, Wossenseged Lemma, Aberham Abere, Yalewayker Tegegne, Dylan R. Pillai.

**Methodology:** Tena Cherkos, Adane Derso, Banchamlak Tegegne, Abebe Birhanu, Kassahun Cherkos, Tegegne Eshetu, Wossenseged Lemma, Aberham Abere, Yalewayker Tegegne, Dylan R. Pillai.

**Resources:** Tena Cherkos.

**Software:** Tena Cherkos, Adane Derso, Abebe Birhanu, Kassahun Cherkos, Tsedenya Gebeyehu, Mulat Yimer, Wossenseged Lemma, Aberham Abere, Yalewayker Tegegne, Dylan R. Pillai.

**Supervision:** Adane Derso, Banchamlak Tegegne, Abebe Birhanu, Kassahun Cherkos, Wossenseged Lemma, Aberham Abere, Yalewayker Tegegne, Dylan R. Pillai.

**Validation:** Tena Cherkos, Adane Derso, Yalewayker Tegegne, Dylan R. Pillai.

**Visualization:** Tena Cherkos, Adane Derso, Banchamlak Tegegne, Wossenseged Lemma, Yalewayker Tegegne, Dylan R. Pillai.

**Writing – original draft:** Tena Cherkos, Adane Derso, Banchamlak Tegegne, Abebe Birhanu, Kassahun Cherkos, Zufan Yiheyis Abreham, Banchayehu Getnet, Tsedenya Gebeyehu, Mulat Yimer, Tegegne Eshetu, Wossenseged Lemma, Aberham Abere, Yalewayker Tegegne, Dylan R. Pillai.

**Writing – review & editing:** Tena Cherkos, Adane Derso, Banchamlak Tegegne, Abebe Birhanu, Kassahun Cherkos, Zufan Yiheyis Abreham, Banchayehu Getnet, Tsedenya Gebeyehu, Mulat Yimer, Tegegne Eshetu, Wossenseged Lemma, Aberham Abere, Yalewayker Tegegne, Dylan R. Pillai.

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
