## [Decision Letter · Decision Letter 0]

22 Aug 2024

PONE-D-24-27859Unveiling the Silent Threat: Investigating Asymptomatic Plasmodium Infections in Gorgora, Ethiopia through Microscopy and Loop-Mediated Isothermal AmplificationPLOS ONE

Dear Dr. Cherkos,

Thank you for submitting your manuscript to PLOS ONE. After careful consideration, we feel that it has merit but does not fully meet PLOS ONE’s publication criteria as it currently stands. Therefore, we invite you to submit a revised version of the manuscript that addresses the points raised during the review process.

We look forward to receiving your revised manuscript.

Kind regards,

Enoch Aninagyei, PhD

Academic Editor

PLOS ONE

https://doi.org/10.1371/journal.pone.0165506

10.1186/s12936-017-1692-4

In your revision ensure you cite all your sources (including your own works), and quote or rephrase any duplicated text outside the methods section. Further consideration is dependent on these concerns being addressed.

Reviewers' comments:

Reviewer's Responses to Questions

**Comments to the Author**

1. Is the manuscript technically sound, and do the data support the conclusions?

Reviewer #1: Partly

Reviewer #2: Partly

Reviewer #3: Yes

2. Has the statistical analysis been performed appropriately and rigorously? 

Reviewer #1: No

Reviewer #2: Yes

Reviewer #3: No

3. Have the authors made all data underlying the findings in their manuscript fully available?

Reviewer #1: Yes

Reviewer #2: Yes

Reviewer #3: Yes

4. Is the manuscript presented in an intelligible fashion and written in standard English?

Reviewer #1: Yes

Reviewer #2: Yes

Reviewer #3: Yes

5. Review Comments to the Author

Reviewer #1: The manuscript by Tena Cherkos and colleagues, describes the use of two-different conventional methods for the detection of Asymptomatic Plasmodium Infections in Gorgora - Ethiopia. Mostly, the manuscript is well written, and the results are presented in a clear manner. However, I consider that the relevance/importance of the obtained results in the present study are not good enough for publication in PLoS One. One of my major concerns is related to the experimental techniques used here and its interpretation, as the authors just performed two well standardised techniques for the detection of Plasmodium parasites. The statistical analysis is poor, and the discussion section is not well supported. It is well known than molecular methods are much more specific and sensitive compared to non-molecular ones. So, the conclusion of the present study could result trivial.

Some other minor concerns that authors should notice are the presented below:

Page 3, lines 83-84. The author states: "The clinical presentation of malaria can vary, exhibiting symptoms ranging from mild and uncomplicated to asymptomatic infections." Reviewer comment: It should be stated that clinical presentation of malaria also includes severe episodes, caused mainly by the parasite Plasmodium falciparum.

Page 3, lines 91-92: The author states: "Therefore, asymptomatic carriers of the Plasmodium parasite can seriously jeopardize the success of attempts to control and eliminate malaria".

Reviewer comment: Did the authors consider administrating the appropriate treatment to the P. falciparum and P. vivax infected population? If not, did the authors recommended to this population to attend the health services?

Page 3, lines 93-94. The author states: "Over the past decades, Ethiopia has committed to intensifying its efforts to eliminate malaria, and significant progress has been made in reducing malaria-related morbidity and mortality [14]"

Reviewer comment: I consider, the reference 14 is nor the most suitable to support the statement. The study in this reference, only includes schoolchild population and the prevalence of API is around 2%, which not seems to be a significant progress in reducing malaria-related morbidity and mortality.

Page 4, lines 107-108. The author states: "Light microscopy is still the gold standard for the diagnosis of malaria because, when properly interpreted, a positive result indicates an active plasmodium parasite infection [25, 26]"

Reviewer comment: I consider that references 25 and 26 are not the most appropriate ones to support the statement, as the main topic of both of them is the use of molecular-based methods for Plasmodium detection. There are available studies discussing specifically the use of light microscopy in Plasmodium infections detection.

Page 5, Study area section.

Reviewer comment: I am not sure if the author should include reference 35 to describe the geographic localisation of the study area. Same for ref 36.

Page 9, Loop-mediated isothermal amplification( LAMP assay) section.

Reviewer comment: This section should be re-written to a less-basic, more technical / scientific manner. I consider that the use of a YouTube video is useless and should be removed.

Page 10, lines 235-236. The author states: "To begin the LAMP assay, 30μl of blood sample and 30 μl 235 of normal saline were transferred into the heating tube containing sodium hydroxide solution."

Reviewer comment: However, in the experimental procedure the author states that blood samples for LAMP studies were dropped into filter papers (DBS).

Page 11, Data processing and analysis section. This section should be re-written and the authors are invited to include more specific details of the statistical analysis they have performed.

Page 14. Discussion section.

Reviewer comment: This section should be re-written and analysed in a deeper and more accurate manner. The vast majority of the references utilised in this section to support the author statements are not appropriate. Most of the references included women in pregnancy to make the comparisons.

Page 15. Lines 320-323. The author states: "In the current study, the rate of malaria positivity using microscopy was 6.1% ( 95%CI:2.5-12.1). This finding was consistent with studies conducted in Sanja, Northwest Ethiopia (6.8%) [47], Rayakobo, Northeast Ethiopia (5.2%) [48], and Boricha District, Sidama Region, Ethiopia ( 4%) [49]."

Reviewer comment: Why is it supposed to exist a "consistency" between the results obtained in the present study and those performed by others? Reference 47 only includes school children in the study. It is not appropriate to compare those results with the obtained in the present study.

Reference 50 describes a study performed in Colombia (South America) using placental blood samples of pregnant women. What is the sense making a comparison between detection of API in the population included in the present study and pregnant women in Colombia (South America)?.

Lines 325-330. The statement should be re-written. The author should not state that "the present study was lower/higher than......". It must be clearly explained.

Reference 53 also includes pregnant women and this reference is not the most suitable to support the statement.

Line 334. The author states: "The present study revealed that LAMP detected the highest number of APIs than microscopy."

Reviewer comment: It is well known than molecular-based methods are most specific / sensitive ones in the detection of Plasmodium parasites.

Lines 349-351. The author states: "Several nucleic acid-based detection methods such as nested PCR, multiplex PCR, and real-time PCR have subsequently been developed and on average showed a detection limit range of around 1–5 P/μL [18, 57, 58]."

Reviewer comment : The references included here do not support the statement.

Line 362. Limitations of the study section.

Reviewer comment: The authors previously known about these limitations. Did the authors consider the possibility of performing the study in another season? a more convenient season? Did the author consider to increase the number of participants in the present study?

Reviewer #2: 1. Quite Small Sample Size for comparison between two method of diagnosis and above all authors are claiming that more robust and costly diagnostic method has edge over convenient and gold standard; considering the fact, author should have bigger sample size and longer time frame rather than using small sample size and shorter time frame. Justify the objective of manuscript based on above comment.

2. Authors have not appropriately rather not even described the method adopted to differentiate the cases of asymptomatic except the verbal recording. Kindly explain/justify the reason in your method on relying on such non-scientific method only.

3.Considering the universal fact that Malaria is the diseases of poor and extensively investigated in field based studies using both active and passive surveillances across the globe and authors have not considered the use of very important field based instant diagnostic tool i.e. RDT Kits in comparison. Kindly justify the reason in your discussion.

4. In absence of any declared funding source, how and why authors designed and implemented such an important qualitative-cum-quantitative study for developing in a lopsided ways in absence of financial constrain. Kindly justify this aspect in your discussion. I strongly feel that very objective required extensive quantitative experimentation/methodology covering sample collection to comparative experimentation limited by the lack of financial support.

Reviewer #3: The author has done good work by showing that LAMP-based assay has the greatest advantage for the detection of APIs, especially in sub-microscopic infections, over microscopic techniques. However, the author should address the below comments before accepting the manuscript for publication.

1) A more detailed discussion of the relationship between sample size and confidence interval is required.

2) The authors have performed a fantastic survey providing detailed contextual information on the case by carrying out extensive research yet, further deeper insights could be strengthened by additionally comparing PCR-based (as performed in the published paper-DOI:10.1186/s12936-021-03923-8) diagnosis data with LAMP based analysis in order to evaluate the data appropriately, providing elaborate implications on their findings thus drawing effective inferences. Thus, additional diagnostic methods, such as PCR (if possible) may enrich the data.

3) The author did not provide any graphical representation of the data. A comparative graphical analysis of the data may be more appropriate.

6. PLOS authors have the option to publish the peer review history of their article (what does this mean? ). If published, this will include your full peer review and any attached files.

**Do you want your identity to be public for this peer review?** For information about this choice, including consent withdrawal, please see our Privacy Policy .

Reviewer #1: No

Reviewer #2: No

Reviewer #3: No

---

## [Author Response · Author response to Decision Letter 0]

25 Oct 2024

Dear Editor,

Thank you for allowing us to revise and it is with great pleasure we resubmit to you a revised version of the manuscript entitled “Unveiling the Silent Threat: Investigating Asymptomatic Plasmodium Infections in Gorgora, Ethiopia through Microscopy and Loop-Mediated Isothermal Amplification” for the Journal: PLOS ONE.

We appreciate your time, valuable comments, and suggestions which will make the manuscript much better. We believe all necessary comments and suggestions are incorporated in the revised manuscript and all changes made are highlighted in yellow color in the revised manuscript and with blue font color in a point-by-point response to reviewers.

We are uploading

(1) Our point-to-point response to the comments below (response to academic editor and reviewer(s), respectively)

(2) A marked-up copy with track changes (Revised Manuscript with highlights)

(3) An unmarked version without track changes (Manuscript)

We appreciate very much your suggestion to improve our article. Below, you can find our response, point by point.

https://journals.plos.org/p1losone/s/file?id=ba62/PLOSOne_formatting_sample_title_authors_affiliations.pdf.

• Response: Thank you. We have adhered to PLOS ONE's style and format requirements in the revised manuscript.

https://doi.org/10.1371/journal.pone.0165506

10.1186/s12936-017-1692-4

• Response: Thank you. We have rephrase and revised based on your comment.

In your revision ensure you cite all your sources (including your own works), and quote or rephrase any duplicated text outside the methods section. Further consideration is dependent on these concerns being addressed.

• Response: Thank you. We have revised the manuscript to address these concerns, rephrasing the duplicated text, and citing sources appropriately.

Response: Thank you. We have incorporate accordingly.

5. Review Comments to the Author

Reviewer #1: The manuscript by Tena Cherkos and colleagues, describes the use of two-different conventional methods for the detection of Asymptomatic Plasmodium Infections in Gorgora - Ethiopia. Mostly, the manuscript is well written, and the results are presented in a clear manner.

Response: Thank you for your positive feedback

However, I consider that the relevance/importance of the obtained results in the present study are not good enough for publication in PLoS One. One of my major concerns is related to the experimental techniques used here and its interpretation, as the authors just performed two well standardised techniques for the detection of Plasmodium parasites. The statistical analysis is poor, and the discussion section is not well supported.

• Response: Thank you for the valuable comment. We revised the method, statistical analysis, and discussion section based on your comments. Please refer accordingly.

It is well known than molecular methods are much more specific and sensitive compared to non-molecular ones. So, the conclusion of the present study could result trivial.

• Response: We appreciate this important comment. Indeed, molecular techniques are more sensitive than microscopy. The objective of this study is to show the performance of the conventional malaria diagnostic method ( microscopy) for the detection of asymptomatic infections particularly submicroscopic level infections since these are hidden reservoirs for continued transmission although it gets less attention. This study may be useful insight for those who control and have a plan to eliminate malaria in a country in which microscopy remains the diagnostic choice. We revised the conclusion accordingly

Some other minor concerns that authors should notice are the presented below:

Page 3, lines 83-84. The author states: "The clinical presentation of malaria can vary, exhibiting symptoms ranging from mild and uncomplicated to asymptomatic infections." Reviewer comment: It should be stated that clinical presentation of malaria also includes severe episodes, caused mainly by the parasite Plasmodium falciparum.

• Response: Thank you. We revised based on your comment, please refer to lines 80-85.

Page 3, lines 91-92: The author states: "Therefore, asymptomatic carriers of the Plasmodium parasite can seriously jeopardize the success of attempts to control and eliminate malaria".

Reviewer comment: Did the authors consider administrating the appropriate treatment to the P. falciparum and P. vivax infected population? If not, did the authors recommended to this population to attend the health services?

Response: Thank you. We put this sentence to show that unlike symptomatic patients asymptomatic plasmodium species carriers do not show symptoms and usually do not seek treatment for their infection. Thus they remain in the community and they are invisible to health facilities. This makes them a hidden reservoir for ongoing malaria transmission and this impedes the efforts to control and eliminate malaria. Active detection of cases might be crucial to scaling up malaria control programs and they should be investigated more. Therefore assessing the prevalence of asymptomatic plasmodium spp infections by using sensitive molecular methods is crucial to report the true burden of the disease and helps to scale up malaria control and elimination programs.

Page 3, lines 93-94. The author states: "Over the past decades, Ethiopia has committed to intensifying its efforts to eliminate malaria, and significant progress has been made in reducing malaria-related morbidity and mortality [14]"

Reviewer comment: I consider, the reference 14 is nor the most suitable to support the statement. The study in this reference only includes schoolchild population and the prevalence of API is around 2%, which not seems to be a significant progress in reducing malaria-related morbidity and mortality.

• Response: Thank you. We revised and replace it with appropriate reference please refer accordingly.

Page 4, lines 107-108. The author states: "Light microscopy is still the gold standard for the diagnosis of malaria because, when properly interpreted, a positive result indicates an active plasmodium parasite infection [25, 26]"

Reviewer comment: I consider that references 25 and 26 are not the most appropriate ones to support the statement, as the main topic of both of them is the use of molecular-based methods for Plasmodium detection. There are available studies discussing specifically the use of light microscopy in Plasmodium infections detection.

• Response: Thank you for the important comment. We have replaced these references with studies discussing specifically the use of light microscopy in Plasmodium infection detection as requested.

Page 5, Study area section.

Reviewer comment: I am not sure if the author should include reference 35 to describe the geographic localisation of the study area. Same for ref 36.

• Response: Thank you. We have checked it carefully. The study area for the present study is Gorgora area which is found in dembia district and it comprises Gorgora town and surrounding rural communities. We use the above two references to show the study area in detail.

Page 9, Loop-mediated isothermal amplification( LAMP assay) section.

Reviewer comment: This section should be re-written to a less-basic, more technical / scientific manner. I consider that the use of a YouTube video is useless and should be removed.

• Response: Thank you we accept the comment. We have removed the YouTube video, rewritten it to a less basic, more technical/scientific manner, and cited it with appropriate references. Please refer to the LAMP method section.

Page 10, lines 235-236. The author states: "To begin the LAMP assay, 30μl of blood sample and 30 μl 235 of normal saline were transferred into the heating tube containing sodium hydroxide solution."

• Reviewer comment: However, in the experimental procedure the author states that blood samples for LAMP studies were dropped into filter papers (DBS).

• Response: Thank you, the comment is valid. First, dried blood spots were punched and eluted with 100 μl nuclease-free water overnight to transfer blood samples from filter papers, and then 30μl of this blood sample was taken for further process. 30μl of blood sample means a sample taken from after liquefying two 3mm DBS by 100 μl nuclease-free water. We have rewritten undoubtedly.

Page 11, Data processing and analysis section. This section should be re-written and the authors are invited to include more specific details of the statistical analysis they have performed.

• Response: Thank you for the valuable comment. We revised it accordingly.

Page 14. Discussion section.

Reviewer comment: This section should be re-written and analysed in a deeper and more accurate manner. The vast majority of the references utilised in this section to support the author statements are not appropriate. Most of the references included women in pregnancy to make the comparisons.

• Response: Thank you. We revised based on your comment. We have removed the references which included women in pregnancy

Page 15. Lines 320-323. The author states: "In the current study, the rate of malaria positivity using microscopy was 6.1% (95%CI: 2.5-12.1). This finding was consistent with studies conducted in Sanja, Northwest Ethiopia (6.8%) [47], Rayakobo, Northeast Ethiopia (5.2%) [48], and Boricha District, Sidama Region, Ethiopia ( 4%) [49]."

Reviewer comment: Why is it supposed to exist a "consistency" between the results obtained in the present study and those performed by others?

• Response: Thank you. We put the consistency to show the reproducibility of our study findings and the possible reasons for the variations. We also put other studies done previously to explain the present findings in accordance with them.

Reference 47 only includes school children in the study. It is not appropriate to compare those results with the obtained in the present study.

• Response: Thank you. We revised based on your comment. We replace reference 47 with a valid reference.

Reference 50 describes a study performed in Colombia (South America) using placental blood samples of pregnant women. What is the sense making a comparison between detection of API in the population included in the present study and pregnant women in Colombia (South America)?

• Response: Thank you. We accept the comment. We have removed ref 50 and replaced it with appropriate reference.

Lines 325-330. The statement should be re-written. The author should not state that "the present study was lower/higher than......". It must be clearly explained.

• Response: Thank you. We state the above phrase "the present study was lower/higher than......". to compare it with previous studies since describing the finding with other studies might help the discussion more stronger and have a scientific concept.

Reference 53 also includes pregnant women and this reference is not the most suitable to support the statement.

• Response: Thank you. We revised based on your comment.

Line 334. The author states: "The present study revealed that LAMP detected the highest number of APIs than microscopy."

Reviewer comment: It is well known than molecular-based methods are most specific / sensitive ones in the detection of Plasmodium parasites.

• Response: Thank you. We put this sentence to show that microscopic diagnosis missed large number of asymptomatic plasmodium infections in this study. Thus submicroscopic plasmodium infections remain undetected and act as a reservoir for forward transmission. To overcome these limitations, the LAMP method which is simple to perform and less costly than PCR can be used as a tool to detect these infections to report the true burden of disease. This study might be a good insight for those who use microscopic diagnosis for surveillance of malaria.

Lines 349-351. The author states: "Several nucleic acid-based detection methods such as nested PCR, multiplex PCR, and real-time PCR have subsequently been developed and on average showed a detection limit range of around 1–5 P/μL [18, 57, 58]."

Reviewer comment: The references included here do not support the statement.

• Response: Thank you. We revised based on your comment and cited it appropriately.

Line 362. Limitations of the study section.

Reviewer comment: The authors previously known about these limitations. Did the authors consider the possibility of performing the study in another season? a more convenient season? Did the author consider to increase the number of participants in the present study?

• Response: Thank you. We have accepted this suggestion and revised it accordingly, please refer to the limitation part of the manuscript.

Reviewer #2: 1. Quite Small Sample Size for comparison between two method of diagnosis and above all authors are claiming that more robust and costly diagnostic method has edge over convenient and gold standard; considering the fact, author should have bigger sample size and longer time frame rather than using small sample size and shorter time frame. Justify the objective of manuscript based on above comment.

Response: Thank you. The objective of this study is to show the performance of conventional malaria diagnostic test for the detection of asymptomatic infections and the use of the LAMP method is to show submicroscopic level infections burden since these are hidden reservoirs for continued transmission though not detected by microscopy. This study may be useful insight for those who control and have a plan to eliminate malaria in a country in which microscopy remains the diagnostic choice.

2. Authors have not appropriately rather not even described the method adopted to differentiate the cases of asymptomatic exc

---

## [Editor Report · Decision Letter 1]

31 Oct 2024

Unveiling the Silent Threat: Investigating Asymptomatic Plasmodium Infections in Gorgora, Ethiopia through Microscopy and Loop-Mediated Isothermal Amplification

PONE-D-24-27859R1

Dear Dr. Cherkos,

We’re pleased to inform you that your manuscript has been judged scientifically suitable for publication and will be formally accepted for publication once it meets all outstanding technical requirements.

Kind regards,

Enoch Aninagyei, PhD

Academic Editor

PLOS ONE
---

## [Editor Report · Acceptance letter]

PONE-D-24-27859R1

PLOS ONE

Dear Dr. Cherkos,

I'm pleased to inform you that your manuscript has been deemed suitable for publication in PLOS ONE. Congratulations! Your manuscript is now being handed over to our production team.

Kind regards,

on behalf of

Dr Enoch Aninagyei

Academic Editor

PLOS ONE